# Lipid-Encapsuled Grape Tannins Prevent Oxidative-Stress-Induced Neuronal Cell Death, Intracellular ROS Accumulation and Inflammation

**DOI:** 10.3390/antiox11101928

**Published:** 2022-09-28

**Authors:** Hugo S. Díaz, Angélica Ríos-Gallardo, Domiziana Ortolani, Esteban Díaz-Jara, María José Flores, Ignacio Vera, Angela Monasterio, Fernando C. Ortiz, Natalia Brossard, Fernando Osorio, Rodrigo Del Río

**Affiliations:** 1Laboratory of Cardiorespiratory Control, Department of Physiology, Faculty of Biological Sciences, Pontificia Universidad Católica de Chile, Santiago 8320000, Chile; 2Departamento de Ciencia y Tecnología de Alimentos, Facultad Tecnológica, Universidad de Santiago de Chile, Santiago 8320000, Chile; 3Mechanisms of Myelin Formation and Repair Laboratory, Instituto de Ciencias Biomédicas, Facultad de Ciencias de la Salud, Universidad Autónoma de Chile, Santiago 8320000, Chile; 4Department of Fruit Production and Enology, School of Agricultural and Forest Sciences, Pontificia Universidad Católica de Chile, Santiago 8320000, Chile; 5Centro de Envejecimiento y Regeneración CARE-UC, Pontificia Universidad Católica de Chile, Santiago 8320000, Chile; 6Centro de Excelencia en Biomedicina de Magallanes (CEBIMA), Universidad de Magallanes, Punta Arenas 6200000, Chile

**Keywords:** oxidative stress, polyphenols, liposomes, neuroprotection, natural products

## Abstract

The central nervous system (CNS) is particularly vulnerable to oxidative stress and inflammation, which affect neuronal function and survival. Nowadays, there is great interest in the development of antioxidant and anti-inflammatory compounds extracted from natural products, as potential strategies to reduce the oxidative/inflammatory environment within the CNS and then preserve neuronal integrity and brain function. However, an important limitation of natural antioxidant formulations (mainly polyphenols) is their reduced in vivo bioavailability. The biological compatible delivery system containing polyphenols may serve as a novel compound for these antioxidant formulations. Accordingly, in the present study, we used liposomes as carriers for grape tannins, and we tested their ability to prevent neuronal oxidative stress and inflammation. Cultured catecholaminergic neurons (CAD) were used to establish the potential of lipid-encapsulated grape tannins (TLS) to prevent neuronal oxidative stress and inflammation following an oxidative insult. TLS rescued cell survival after H_2_O_2_ treatment (59.4 ± 8.8% vs. 90.4 ± 5.6% H_2_O_2_ vs. TLS+ H_2_O_2_; *p* < 0.05) and reduced intracellular ROS levels by ~38% (*p* < 0.05), despite displaying negligible antioxidant activity in solution. Additionally, TLS treatment dramatically reduced proinflammatory cytokines’ mRNA expression after H_2_O_2_ treatment (TNF-α: 400.3 ± 1.7 vs. 7.9 ± 1.9-fold; IL-1β: 423.4 ± 1.3 vs. 12.7 ± 2.6-fold; *p* < 0.05; H_2_O_2_ vs. TLS+ H_2_O_2_, respectively), without affecting pro/antioxidant biomarker expression, suggesting that liposomes efficiently delivered tannins inside neurons and promoted cell survival. In conclusion, we propose that lipid-encapsulated grape tannins could be an efficient tool to promote antioxidant/inflammatory cell defense.

## 1. Introduction

During life, the central nervous system (CNS) is constantly exposed to reactive oxygen species (ROS) and other free radicals emerging from many sources, including exposure to oxidizing agents (smoke, ionizing radiations, toxic agents) and body-borne oxidants, such as catecholamine metabolites, mitochondrial side-metabolites and polyunsaturated fatty acids, which are abundantly present in the CNS [1,2]. Given the limited regenerative capacity of neurons, the CNS is particularly vulnerable to ROS; therefore, to maintain a correct balance between antioxidants and ROS is fundamental for preventing oxidative stress and CNS inflammation, which can result in neuronal damage or death [3,4,5]. In fact, it is widely accepted that oxidative stress and neuronal inflammation play a major role in several chronic CNS neurodegenerative pathologies, including Parkinson’s disease (PD), Alzheimer’s disease (AD), Huntington’s disease (HD) and amyotrophic lateral sclerosis (ALS) [6,7,8,9,10]. Among ROS, hydrogen peroxide (H_2_O_2_) emerges as a the major redox signaling molecule, given it is produced in all cell types by multiple sources (mitochondria, NAD(P) oxidase, and several cytosolic and microsomal peroxidases [11]), as well as its capacity to activate the major master switches related to inflammation and cell death after supraphysiological (>100 nM) production, including NF-κB (nuclear factor kappa B), AP-1 (Activator Protein-1) and TLRs (Toll-like Receptors) [2,11,12], major inflammatory pathways which mediate immune cell recruitment and damage signaling [13], resulting in neuronal cell death and neurodegeneration [14,15].

Lifestyle and diet strongly affect redox balance in the genesis and progression of neurodegenerative diseases [16], and there is a growing interest on natural antioxidants, particularly in polyphenol-rich foods for the prevention of CNS oxidative stress and neurodegeneration [17,18,19,20]. In fact, there is strong evidence showing neuroprotective functions of diet polyphenols against neuroinflammation, preserving cognitive functions such as memory and learning [16,19,20,21]. In this regard, grape seeds and skin are considered as valuable and eco-friendly sources of bioactive compounds with neuroprotective potential, as their extracts show neuroprotective activities against neurodegeneration, including AD and cerebral ischemia [22,23] by several mechanisms including ROS scavenging, prevention of synaptic dysfunction and neuronal cell death and inhibition of neuroinflammation [23]. Polyphenols can exert their antioxidant capacity through different mechanisms, including direct ROS scavenging, chelation of pro-oxidant metal ions, inhibition of enzymes involved in ROS production, upregulation of antioxidant enzymes and downregulation of pro-inflammatory genes [18,23,24,25]. Considering the important role of oxidative stress and neuroinflammation in the progression of neurodegenerative diseases, as well as the high potential of polyphenols to mitigate these responses by several pathways, they emerge as suitable candidates for preserving CNS function and preventing/diminishing neurodegeneration.

Polyphenols comprise over 8000 bioactive compounds, characterized by aromatic rings with one or more hydroxyl groups [26], being majorly classified into groups and subgroups based on the numbers of phenolic rings and structural elements attached to the rings whose arrangement, configuration, substitution and the number of hydroxyl groups influence the antioxidant activity such as the radical scavenging activity and or metal ion chelation ability [26]. Tannins are grape-derived condensed polyphenols with high antioxidant capacity, and they are very important for red wine quality due to their role in astringency and long-term color stability [27,28]. According to their chemical structure and antioxidant properties, it is possible to hypothesize that grape tannins may help protect the CNS (i.e., degeneration) or mitigate the damage caused by oxidative insults [18]. In support of this notion, there is evidence that daily oral administration of high doses of tannic acid diminishes neuroinflammation and cognitive impairment in in vivo models of AD [29,30], showing a high potential of grape tannins as neuroprotective agents. However, a major problem limiting the therapeutic potential of grape or any other polyphenol mixture resides in their in vivo bioavailability [31], which depends on many factors such as intake rate and intestinal absorption [16]. Indeed, even though polyphenols are abundant in many daily foods, they are poorly absorbed, the outcome being poor levels in human circulation [31]. A method that can manage the reduced bioavailability of polyphenols is the use of molecular carriers that allow proper delivery to living cells. In this regard, liposomes have been previously contemplated as one of the most promising vehicles for drug delivery since they facilitate absorption by fusion with the plasma membrane of live cells of human/biological tissues [32]. Liposomes are self-organizing colloidal particles that constitute of one or more lipid bilayer membranes, which surround an aqueous compartment that can be utilized to encapsulate both hydrophilic, hydrophobic and amphiphilic compounds [33]. The use of liposomes provides several advantages such as low toxicity, biocompatibility, lower clearance rates, the ability to target specific tissues and the controlled release of drugs [32,34]. Importantly, liposome structure can also serve to deliver drugs into the brain since they are able to cross the blood–brain barrier (BBB), suggesting a strong potential of liposomes for the delivery of polyphenolic mixtures to the brain, which then act as neuroprotectors. Indeed, previous studies have demonstrated that tannic-acid-charged liposomes can reduce Tau aggregation in human neuroblastoma cells, and have shown to significantly reduce intracellular ROS production upon oxidative stress induction [35,36]. In addition, recent reports have shown that orally administered tannins are capable to reduce brain neuroinflammation [29], supporting a high potential of grape tannins as novel protective agents against neuroinflammation. However, limited information is available regarding the effects of lipid-encapsulated grape tannins on in vitro and in vivo models of neuroinflammation. In this study, we used liposomes to encapsulate/package grape polyphenols to tackle their poor bioavailability and low stability. We thus tested the neuroprotective capacity of lipid-encapsulated grape tannins formulation against oxidative stress and neuroinflammation in a catecholaminergic neuronal cell line (CAD) [37], by evaluating their effects on cell viability under oxidative stress conditions, intracellular ROS production and oxidative/inflammatory gene expression.

## 2. Materials and Methods

### 2.1. Liposome Preparation and Tannins Encapsulation

Tannin-containing liposomal nanosuspensions (TLS) were prepared using the heating/homogenization method with lamellarity and size reduction by ultrasound cycles [38]. To prepare 100 mL of TLS [1 mg/mL], 0.1 g of condensed tannins obtained from green grape seeds (Sauvignon blanc variety) were dissolved in 20 mL of a solution (50:50% *v*/*v*) of ethanol-citrate buffer (0.1 M at pH 3) at 70 ± 1 °C and 700 rpm, respectively. Once dissolved, 1 g of phosphatidylcholine (PC) was incorporated and stirred at 700 rpm without temperature for 5 min. The suspension was then heated at 80 °C for 1 h in a thermoregulated bath. After this time, 0.76 g of glycerol was added as a lipoprotectant [39] dissolved in 40 mL of citrate buffer (0.1 M at pH 3.0), and heated again at 80 °C for 1 h. After this second heating period, the remaining 40 mL of buffer was added to complete the volume and 5 vortex cycles, and 20 ultrasound cycles were applied, respectively, using an ultrasonic cell disruptor (HIELSCHER UP100H, Teltow, Germany, max. 100 W) with a sonotrode. MS7 Micro tip 7 (7 mm diameter, 120 mm length, 130 W/cm^2^ acoustic power density) working at 90% amplitude and 22.5 Hz. Finally, TLS were purified by centrifugation at 2500× *g* for 15 min at 15 °C and redispersed in MilliQ water. PC was obtained by purification of raw food-grade soy lecithin following the procedure described by López-Polo et al. to obtain saturated lipids [40]. Crude soy lecithin (10 g) was dissolved in 50 mL of ethyl acetate at 20 °C. Distilled water (2 mL) was then slowly added with manual stirring, resulting in the formation of two phases. The lower phase was separated and dispersed in 30 mL of acetone, forming clusters that were crushed with a glass rod. Then, the acetone was separated by decantation and a new aliquot (30 mL) of acetone was added, repeating the trituration process. The precipitate was vacuum filtered and dried in a desiccator at 20 °C for 48 h to finally obtain soy phospholipids (dipalmitoyl lecithin (1,2-dipalmitoyl-sn-glycerol-3-phosphocholine, DPPC, M.W.:734.05 g/mol, purity ≥ 99%). In total, 1 mg/mL of condensed tannin was encapsulated; this concentration was preliminarily determined with a multilevel factorial statistical design, which contemplated concentrations from 0.5 to 5 mg/mL of tannins. For determination of mean degree of polymerization, a suspension of condensed tannins (TS) was used as control. Tannic acid (hydrolysable) was used as a standard (Sigma, St. Louis, MI, USA; 403040) for those monomers that have remained free after vesicle formation with the ultrasonic disruptor, to determine encapsulation efficiency and loading capacity, by interpolation by spectrophotometry using tannic acid as standard at 280 nm absorbance.

Encapsulation efficiency and drug loading efficiency were calculated using the formulas:
(1)$$\mathrm{Encapsulation}\;\mathrm{efficiency}\;\left(\%\right)=\frac{\mathrm{weight}\;\mathrm{of}\;\tan\mathrm{nins}\;\mathrm{into}\;\mathrm{nanoparticle}}{\mathrm{initial}\;\mathrm{weight}\;\mathrm{of}\;\tan\mathrm{nins}}\times 100=\frac{C\times V-C^{\prime }\times V'}{C\times V}$$
$$\mathrm{Loading}\;\mathrm{efficiency}\;\left(\%\right)=\frac{m}{M}\times 100=\frac{C\times V-C^{\prime }\times V'}{M}$$

With *C* = dosage concentration, *V* = volume of administration, *C**′* = drug concentration in the separation solution after drug loading, *V**′* = total volume of liposome suspension, *m* = weight of tannins entrapped into liposomes, and *M* = weight of product (liposome + loaded tannins). *C**′* was determined by spectrophotometry by using standard curves of tannic acid standard at 280 nm absorbance (y = 0.021x − 0.114, r^2^ = 0.955, *p* < 0.001). *M* was determined by weighting the dehydrated nanoparticles after freeze-drying known aliquots of the dispersion

### 2.2. Mean Particle Size (MPS), Polydispersity Index (PDI), and z-Potential (ξ)

The MPS, PDI and ξ analyses for each sample were performed using dynamic light scattering (DLS) with a measurement angle of 173° Backscater and a mean and phospholipid refractive index of 1.330 and 1.334, respectively. (Zetasizer Nano ZS, Malvern Instruments, Malvern, UK).

### 2.3. Proximal Composition

Proximal composition (see Supplemental Appendix A) of tannins (TS) and nanoliposomes that encapsulated grape seed tannins (TLS) were carried out under the standards 925.45 for moisture; 923.03 for ashes; 990.03 for proteins; 996.06 for lipids of the A.O.A.C., 2016.

### 2.4. Quantification of Total Phenols

For the quantification of total phenols, expressed as gallic acid [μg/mL], the methodology described by Cano et al. (2020) was used with some modifications [41]. Briefly, 1 mL of 1 mg/mL TS or TLS (redispersed after centrifugation) was added to a 15 mL conical tube. A total amount of 200 μL of Folin–Ciocalteu reagent was added and allowed to stand for 3 min. After this time, 1.6 mL of 7.5% sodium carbonate solution was added and it was left to react for 1 h in the absence of light. Absorbance data were recorded at 760 nm for later quantification.

### 2.5. HPLC-UV

The identification of the compounds present in TS and TLS were carried out by means of depolymerization of tannins in acidic methanol and in the presence of toluene-α-thiol or cysteamine hydrochloride (acid thiolysis) and subsequent HPLC-UV analysis using a wavelength of 280 nm [42]. To extract the condensed tannins from the nanoliposomes, they were centrifuged at 9000 rpm for 1 h at 20 °C. In total, 200 μL of the supernatant was transferred to a microcentrifuge tube (2 mL) and mixed with 200 μL of thiolysis medium (Cysteine-Hydrochloric Acid-Methanol). Subsequently, the microcentrifuge tube was heated in a water bath at 65 °C for 1 h to carry out thiolysis. During thiolysis, the monomeric units of flavan-3-ol are released from the condensed tannins in their native state (terminal units) as well as their respective thioethers (extending units). After the thiolysis time, 1 mL of deionized water was added to stop the reaction [42]. In total, 25 μL of the sample (thiolyzed extract, non-thiolyzed extract, calibration standard) was injected onto the HPLC-UV column at 280 nm. Prior to injection, samples were conditioned to room temperature (20 °C). The mobile phase gradient described by Bianchi et al. [42] was applied at a flow equal to 1 mL/min. The composition of Phase A was 0.1% trifluoroacetic acid (TFA) in deionized water and Phase B was 0.08% TFA in a 4/1 ratio of acetonitrile/deionized water.

### 2.6. Trolox Equivalent Antioxidant Capacity Assay

The Antioxidant Assay Kit (Cayman Chemical, Ann Arbor, MI, USA, 709001) was used to measure antioxidant capacity of TLS and TS preparations (redispersed to 1 mg/mL after centrifugation), according to the method described by Compaoré et al., 2016 [43], based on the capacity of the sample to prevent the oxidation of ABTS (2,2′-azino-di-[3-ethylbenzthiazoline sulphonate]) to ABTS•+. ABTS assay was performed according to the manufacturer instructions with minimal modifications. Briefly, 10 μL of suspension or Trolox standard was added to 200 μL of diluted ABTS solution, with incubation in the dark for 5 min. The absorbance was read at 620 nm, with a microplate reader (BioTek Instruments, New York, NY, USA). Trolox was used to generate the standard curve (y = −0.552x + 0.307, r^2^ = 0.998, *p* < 0.001), and the results were expressed in micromole of Trolox equivalents (TE) per mL of the initial preparation (µmol TE/mL).

### 2.7. Cell Culture

Mouse (B6/D2 F1 hybrid) catecholaminergic neuronal cell line (CAD) was purchased from the European Collection of Authenticated Cells lines (Cat No. 08100805). Cells were grown in DMEM/F-12 medium (Gibco, Waltham, MA, USA, 11330-032), supplemented with 8% FBS (Gibco, Waltham, MA, USA, 10437-028) and 1% penicillin-streptomycin (10,000 U/mL penicillin G sodium and 10,000 μg/mL streptomycin sulfate, (Sigma, St. Louis, MO, USA, P4333) on standard 100 mm cell culture dishes at 37 °C in a humidified 5% CO_2_ incubator. CAD cells were passaged every 3–4 days by pipetting cells from a confluent plate and triturating them in 1 mL of fresh medium. Cells were replated at a 1:10 dilution. For in vitro experiments, cells were differentiated to neuronal phenotype according to previously reported procedures [37]. To induce differentiation, CAD cells were plated in serum-containing DF12 medium, and then switched to serum-free medium (SFM), (60% confluency). SFM contained 20 μg/mL transferrin human (Sigma, St. Louis, MO, USA, T8158) and 50 ng/mL sodium selenite (Sigma, St. Louis, MO, USA, S5261) in DF12 medium. Differentiated CAD cells were grown for at least 5–7 days in SFM before they were used for the experiments.

### 2.8. Cell Viability Assay

In total, 40,000 CAD cells were seeded in 6-well plates and differentiated for 7 days. After differentiation, cells were incubated with charged TLS, empty LS (1 mg/mL) and TS (0,1 mg/mL) for 6 h at 37 °C y 5% CO_2_. After the incubation, the toxicity of H_2_O_2_ on cultured CAD cells was assessed by the trypan blue exclusion method (Figure 1A). After exposure to 200 µM H_2_O_2_ or SFM (control plates) for 24 h at 37 °C 5% CO_2_, cells were immediately stained with 1.5% Trypan blue for 10 min at room temperature. Cells were then examined by light microscopy (20x), counted in quadruplicate in the Neubauer chamber for determining the viability.

### 2.9. Intracellular ROS Assay

In total, 2500 CAD cells were plated in 35 mm dishes, differentiated for 7 days and first incubated with TS (0,1 mg/mL), TLS or LS (1 mg/mL) for 1 h at 37° and 5% CO_2_. CellROX^®^ Deep Red Reagent^®^ (Thermo Fisher Scientific, Waltham, MA, USA, C10422) was then added in each well at 5µM for 30 min in the same experimental conditions. Then, 10 mM H_2_0_2_ was added, and a time lapse (30 min, 2 Hz) was acquired by confocal microscopy (Zeiss 710, 10X objective, Jena, Germany) at 644/665 nm Ex/Em, respectively (Figure 2A).

### 2.10. RNA Isolation, cDNA Synthesis and RT-qPCR Analysis of Neuroinflammatory Biomarkers

In total, 100 mm culture dishes containing 100,000 differentiated CAD cells were used for molecular biology experiments. Briefly, cells were treated with TS, LS, and TLS with or without 200 µM H_2_O_2_ for 24 h (Figure 3A). Immediately after, cells were washed twice with ice-cold 1X PBS for stopping stimuli and RNA isolation was immediately performed using TRIzol reagent^TM^ (Invitrogen 15596026) according to the manufacturer instructions. RNA samples were stored at −80 °C until quantification and cDNA synthesis by reverse transcription. RNA was quantified in Take3^TM^ plates in Epoch microplate reader (BioTek Instruments, New York, NY, USA). RNA purity (260/280 ratio) was 1.84 ± 0.06. cDNA synthesis was performed immediately after using the iSCRIPT kit (Bio-Rad, Hercules, CA, USA, 1708891) from 1 µg of RNA per reaction, according to the manufacturer instructions using the next thermal profile (1 cycle): 25 °C 5 min for priming, 46 °C 20 min for RT, 1 min 95 °C for RT inactivation, followed by immediate incubation in ice for at least 30 min before storage. cDNAs were stored at −20 °C until qPCR experiments. qPCR reactions were performed in AriaMx Real-time PCR Platform (Agilent, Santa Clara, CA, USA) using SYBR-Green chemistry (Kapa SYBR^®^ Fast Universal 2X Mastermix, Sigma KK4601, St. Louis, MO, USA) and 200 nM standard-desalted primers (Integrated DNA Technologies, IA, USA). Primers sequences used in this study were: TGF-β1 F: 5′-GTGCGGCAGCTGTACATTGACTTT-3′, R: 5′-TGTACTGTGTGTCCAGGCTCCAAA-3′; TNF-α F: 5′-AGCCCCCAGTCTGTATCCTT-3′, R: 5′-CTCCCTTTGCAGAACTCAGG-3′; IL-1β F: 5′-TGTGGCTGTGGAGAAGCTGT-3′, R: 5′-CAGCTCATATGGGTCCGA-3′; Cu-Zn SOD F: 5′-CCACCATGTTTCTTAGAGTGAGG-3′, R: 5′-AACCAGTTGTGTTGTCAGGAC-3′; Mn SOD F: 5′-CCAAGGGAGATGTTACAACTCAG-3′, R: 5′-GGGCTCAGGTTTGTCCAGAA-3′; nNOS F: 5′-GGTCTTCGGGTGTCGACAA-3′, R: 5′-GAGTAGGCAGTGTACAGCTCTCTGA-3′; and 18S used as internal standard, F: 5′-GTAACCCGTTGAACCCCATT-3′, R: 5′-CCATCCAATCGGTAGTAGGC-3′. PCR reactions were performed in a final volume of 10 µL using 1:10 diluted cDNA samples using the next thermal profile: initial enzyme activation by 3 min at 95 °C followed by 45 cycles consisting of: 3 s at 95 °C (denaturation) and 60 °C for 30 s (annealing/extension/acquisition), followed by a melting curve from 72 to 95 °C (0.3 °C/read). qPCR reactions were performed in duplicate for each sample, using the 2^ΔΔCT^ method for quantifications as described previously [44,45], using the geometric mean of CT values of control samples as internal calibrators for each target. We also tested the constitutive expression of 18S among all the experimental groups by using 2^−ΔCT^ comparisons between calibrator and treated samples (Appendix A). CT values over 35 were considered insignificant and were excluded for RT-qPCR quantifications.

### 2.11. Statistical Analysis

The data in the tables are presented as mean ± standard deviation (S.D.), and for violin plots as median and quartiles. Normal distribution of the data was assessed using a Shapiro–Wilk test. Comparisons were performed through Student’s *t*-test, one-way ANOVA test followed by Bonferroni’s post hoc test, and two-way ANOVA with repeated measurements, according to the data structure. The level of significance was defined as *p* < 0.05. All the statistical analysis was performed with GraphPad Prism 9.0 software (Dotmatics, Boston, MA, USA).

## 3. Results

### 3.1. Lipid-Encapsulated Tannins Characterization

TS and TLS samples presented low protein content 0.08 [g/100 g], attributed to the interaction of tannins with proteins. Regarding lipids content, TLS reported a high value of 0.18 [g/100 g], which is mainly attributed to the addition of soy phosphatidylcholine. The energy contribution of TS (11.39) and TLS (12.04 kcal/100 g) was lower than the value reported by Czaplicka et al. (2016) [46] for fresh grapes 33 kcal/100 g (Table 1). The encapsulation efficiency was determined by spectrophotometry using tannic acid as standard at 280 nm absorbance. The encapsulation efficiency of TLS was 91.0 ± 5.1 %, and the loading efficiency was 53.0 ± 3.0%. ABTS assay revealed that TLS showed negligible antioxidant activity compared to TS (Table 1, 9.10 ± 6.75 vs. 2017.51 ± 238.57 µM TEAC, *p* < 0.05; Table 1). In fact, in three of five independent assays, ABTS antioxidant activity was equal to 0 µM TEAC. As expected, total polyphenols content (TPC) was significantly higher in TS compared to TLS (78.34 ± 0.12 vs. 30.23 ± 0.14 GAE µg/mL, TS vs. TLS, *p* < 0.05) per mL (Table 1). These results could be attributable to the w/w tannins ratio present in TS and TLS formulations, since both TPC and ABTS assays were performed using 1 mg/mL of redispersed particles after centrifugation. Considering that TS consists of free tannin suspension, whereas TLS loading efficiency was 53%, lower antioxidant activity of TLS was expectable, since tannic acid concentrations per mL of TLS are two-fold lower compared with TS.

Appendix A shows the concentrations (mg/L) obtained by HPLC-UV at 280 nm of the different monomeric units of flavan-3-ol of the condensed tannins in their native state (terminal units) as well as their respective thioethers (units of extension) such as gallic acid, catechin, catechin-cysteine, epicatechin, epicatechin-cys, catechin gallate, epicatechin gallate, epitecatechin gallate cys and procyanidin B1(cis-trans) in the samples of suspended tannins (TS) and encapsulating nanoliposomes grape seed tannins (TLS), with catechin being the most abundant component in both samples, followed by epicatechin and gallic acid (Appendix A).

### 3.2. Tannins and Charged Liposomes Prevent Hydrogen Peroxide-Induced Cell Death

We analysed the effects of empty lyposomes (LS), TS and TLS on cell viability in differentiated CAD cells treated with or without hydrogen peroxide, in order to explore the capacity of nanoliposomes and free tannin suspensions to prevent ROS-induced neuronal cell death. For this, CAD cells were preincubated with 0.1 mg/mL LS, TLS or TS for 30 min before adding 200 µM H_2_O_2_

We found that 200 µM H_2_O_2_ treatment induced cell death compared with the control (100.00 ± 7.01 vs. 59.40 ± 17.62% of survival, control vs. H_2_O_2_, *p* < 0.05; Figure 1). Interestingly, both TS and TLS prevented hydrogen peroxide-induced cell death in CAD cells when compared to the H_2_O_2_ treated control (59.40 ± 17.62% vs. 90.38 ± 11.17 and 95.38 ± 10.32% of survival; H_2_O_2_ vs. H_2_O_2_ + TLS and H_2_O_2_ + TS, respectively; *p* < 0.05), despite TLS showed negligible antioxidant activity by the ABTS method and significantly lower polyphenols content expresed as gallic acid equivalents (Table 1). These data suggest that TLS efficiently delivered tannins inside the cells before H_2_O_2_ exposure and then prevented hydrogen-peroxide-induced neuronal cell death since empty liposomes were unable to prevent cell death after H_2_O_2_ treatment when compared with charged liposomes (48.72 ± 21.14 vs. 90.38 ± 11.17% cell survival, H_2_O_2_ + LS vs. H_2_O_2_ + TLS, respectively; *p* < 0.05 Figure 1). Neither LS, TLS or TS alone affected cell survival (100.00 ± 7.01 vs. 98.02 ± 3.84 vs. 96.58 ± 3.97 vs. 96.15 ± 9.42; Control vs. LS vs. TLS vs. TS, respectively).

### 3.3. Tannins and Charged Liposomes Prevented Intracellular ROS Production after Hydrogen Peroxide Trearment

To confirm if TLS exerted their neuroprotective effects by efficient intracellular ROS scavenging, we pre-stimulated CAD cells with TLS (0.1 mg/mL final concentration) and TS (0.1 mg/mL) during ROS intracellular measurements using live-cell imaging [47]. After acute stimulation with 10 mM H_2_O_2_, sustained increases in intracellular ROS were observed in control and LS (0.1 mg/mL) treated cells (Figure 2B), confirming that liposomes, per se, did not exert neuroprotective effects against oxidative stress. Quantitative analysis of time-lapse images showed that treatment with TLS and TS decreases intracellular ROS levels in CAD cells exposed to H_2_O_2_ when compared with controls (Figure 2B; *p* < 0.05). Indeed, CellROX^®^ fluorescence (Figure 2C) was markedly reduced by TLS in CAD cells following H_2_O_2_ exposure (AUC 4.46 × 10^6^ ± 4.80 × 10^5^ vs. 2.76 × 10^6^ ± 2.22 × 10^5^ arbitrary units; control vs. TLS, respectively, *p* < 0.05). As expected, ROS levels between TLS and TS were comparable (AUC 2.76 × 10^6^ ± 2.22 × 10^5^ v.s. 2.21 × 10^6^ ± 3.68 × 10^5^ arbitrary units; TLS vs. TS, respectively).

### 3.4. Lipid-Encapsulated Grape Tannins Protects Neurons against ROS-Induced Neuroinflammation

We analysed the capacity of TLS (0.1 mg/mL concentration in plate) to prevent oxidative stress-induced neuroinflammation as well as to induce the expression of antioxidant gene defense in neurons. For that, we treated CAD cells with 200 µM H_2_O_2_ with or without TLS for 24 h and we then analysed pro/anti-inflammatory cytokines expression and antioxidant enzymes by RT-qPCR. As shown in Figure 3B,C, H_2_O_2_ resulted in marked increases in proinflammatory cytokines expression and reductions in antioxidant enzyme expression. Importantly, TLS treatment dramatically reduced the mRNA expression of two major proinflammatory cytokines after H_2_O_2_ treatment (tumor necrosis factor alpha, TNF-α: 400.3 ± 1.7 vs. 7.9 ± 1.9-fold; and interleukin-1 beta, IL-1β: 423.4 ± 1.3 vs. 12.7 ± 2.6-fold; *p* < 0.05; H_2_O_2_ vs. TLS+ H_2_O_2_, respectively), suggesting a neuroprotective role of TLS by reducing neuronal inflammation. H_2_O_2_ treatment resulted in reduced expression of two main antioxidant enzymes, soluble superoxide dismutase (CuZn-SOD, 1.03 ± 0.12 vs. 0.28 ± 0.08-fold, Control vs. H_2_O_2_, *p* < 0.05) and mitochondrial superoxide dismutase (Mn-SOD, 1.59 ± 0.21 vs. 0.10 ± 0.06-fold, Control vs. H_2_O_2_, *p* < 0.05), and, surprisingly, neither TLS nor TS were able to restore their levels to control condition (CuZn-SOD, 0.28 ± 0.08 vs. 0.12 ± 0.04 and 0.26 ± 0.03-fold; H_2_O_2_ vs. H_2_O_2_ + TLS and. H_2_O_2_ + TS, respectively, n.s.; Mn-SOD, 0.10 ± 0.06 vs. 0.63 ± 0.22 and 0.44 ± 0.21-fold; H_2_O_2_ vs. H_2_O_2_ + TLS and. H_2_O_2_ + TS, respectively, n.s. Appendix A). Neither H_2_O_2_ nor TLS affected neuronal nitric oxide synthase expression (Figure 3B). In addition, we found that transforming growth factor beta-1 (TGF-β1) was also increased by H_2_O_2_ treatment and this was prevented by TLS treatment (1.03 ± 0.10 vs. 16.21 ± 1.65 vs. 0.34 ± 0.14-fold; Control vs. H_2_O_2_ vs. H_2_O_2_ + TLS, respectively, *p* < 0.05 among groups). LS (1 mg/mL) did not restore TNF-α, IL-1β, TGF-β1 and SOD expression levels in H_2_O_2_-treated CAD cells (Appendix A). Regarding to free grape seeds tannins suspensions, TS-treated (0.1 mg/mL) CAD cells displayed a non-significant trend to restore pro-oxidant and pro-inflammatory biomarker mRNA expression in response to H_2_O_2_, with the exception of IL-1β, on which H_2_O_2_ + TS showed a significantly higher anti-inflammatory activity compared with H_2_O_2_ + TLS (2.1 ± 0.3 vs. 12.7 ± 2.6-fold expression vs. control, respectively; *p* < 0.05; Supplementary Figure 1). Therefore, lipid-encapsulated grape seed tannin treatment reduced H_2_O_2_-induced proinflammatory gene expression without affecting main antioxidant genes. Considering that TLS prevented neuronal cell death (Figure 1), intracellular ROS formation (Figure 2) and classic proinflammatory cytokine mRNA expression (Figure 3) in response to H_2_O_2_, our results suggest that TLS may exerted neuroprotection by these mechanisms.

## 4. Discussion

The main findings of our work demonstrate that both grape tannins suspension (TS) as well as lipid-encapsulated grape tannins (TLS) exerted neuroprotective effects, preventing ROS-induced neuronal cell death, reducing intracellular ROS and diminishing proinflammatory gene expression, with no effect on main antioxidant enzyme expression, suggesting that the lipid encapsulation of tannins resulted in the effective delivery of grape seed polyphenols to neurons.

Tannic acid is one main polyphenol present in grape seeds and skin [27,28] and exerts strong antioxidant properties [48]. Therefore, its utilization as a natural source of antioxidants is a potential alternative for the development of novel neuroprotective drugs. However, the limited bioavailability of dietary polyphenols such as tannic acid mine its potential as a neuroprotective agent [31]. However, lipid encapsulation has been proposed as a feasible solution to cope with low bioavailability in vivo [32]. Indeed, our results showing that TLS antioxidant activity was negligible compared with TS (Table 1), and that TLS significantly (i) reduced intracellular ROS levels and (ii) reduced pro-inflammatory cytokines expression in CAD cells, strongly support the finding that lipid encapsulation is an efficient way to deliver grape tannins to neurons, acting as neuroprotectors by directly scavenging ROS as well as reducing neuronal inflammation. Indeed, empty liposome (LS) completely failed to protect cells against H_2_O_2_-induced cell mortality and intracellular ROS. Remarkably, we found that TLS mimicked the neuroprotective effects of free tannins suspension (TS). Considering that loading efficiency was ~50% and encapsulation efficiency ~90%, approximate tannins concentration in 0.1 mg/mL of TLS is ~0.05 mg/mL, TLS showed 2-fold more in vivo antioxidant capacity compared to TS (0.1 mg/mL). Therefore, given that TLS mimicked TS neuroprotective effects (even at lower theoretical concentrations than free tannins suspensions) and that LS failed to protect neurons against oxidative stress, we believe that our results strongly suggest that observed effects of grape-tannin-charged liposomes are attributable to their cargo.

Neuroinflammation is a complex orchestrated process that comprises all CNS cells, including astrocytes, microglia, neurons and infiltrated leukocytes [4,49]. Previous reports showed that catecholaminergic neurons respond to proinflammatory stimuli by producing TNF-α, IL-1β, IL-6, among other inflammatory biomarkers [50,51]. The later support the notion that neurons actively participate in the neuroinflammatory response. In the present study, we found that ROS accumulation elicit an inflammatory response in cultured neurons in vitro and that lipid-encapsulated grape tannins are an effective mean to reduced ROS-induced neuroinflammation. Since we did not perform experiments in whole animals, caution is needed when trying to translate our findings into animal physiology. Further experiments are needed to fully determine the translational potential of lipid-encapsulated grape tannins on CNS inflammation in animals. Due to the nature of our study, we did not investigate the precise mechanism(s) by which TLS exerts neuroprotection. However, it has been shown that both NF-κB and AP-1 transcription factors are involved in the cellular responses following hydrogen peroxide-induced neuroinflammation in both physiological and pathological conditions [2,11,12,13,14,15]. Despite the fact that we did not directly assess these signaling pathways, TNF-α and IL-1β (which we found to be elevated after H_2_O_2_ treatment and reduced by TLS) are classic downstream targets of the NF-κB/AP-1 signaling pathway in neurons and glial cells [4,23,29,52], and there is evidence showing that pro-oxidant stimuli activates NF-κB and AP-1 in catecholaminergic neurons being the outcome of an increased expression of TNF-α and IL-1β [37]. Therefore, it is highly likely that the beneficial effects associated with TLS are partly linked to inhibition of the NF-κB/AP-1 signaling pathway in CAD neurons. Future studies are needed to fully determine the exact molecular mechanism(s) by which TLS offers neuroprotection.

Our results showed that lipid encapsulation of tannins could be an alternative strategy to prevent some of the pathological features of neurodegenerative diseases (i.e., AD, HD, PD, ALS) since they effectively reduced neuronal oxidative stress and inflammation, two main common hallmarks of these pathological states [6,7,8,9,10]. A recent report showed that periodic administration of high doses of tannic acid (not condensed) prevents cognitive impairment in an AD-like model induced by lipopolysaccharide intraperitoneal injections (the major activator of the proinflammatory TLR4 signaling pathway) [29]. This study confirms a major role of inflammation and oxidative stress in the genesis and progression of neurodegenerative disease, as well as the potential of dietary polyphenols to mitigate neuroinflammation and symptoms of neurodegenerative disease [29]. Interestingly, in the same study, authors were able to find that high doses of tannic acid (60 mg/kg) were effective in reducing LPS-induced neuroinflammation and cognitive impairment [29]. However, this high dosage of tannic acid is unlikely to be translated into human studies due to the well-known low tolerability of high dose tannic acid. Then, a carrier-based strategy may result in lower dosages without comprising tannins therapeutic potential. Given their capacity to easily cross the BBB and the chemical protection of their cargo, liposomes-encapsulated tannins rise as a promising tool to afford neuroprotection in vivo [32,34].

## 5. Conclusions

In summary, our study provided the first evidence that lipid-encapsulated grape tannins are an effective means to control oxidative stress and inflammation in neurons following an oxidative insult. Then, lipid-encapsulated grape tannins could offer a novel strategy to confer neuroprotection.

## Figures and Tables

**Figure 1 antioxidants-11-01928-f001:**
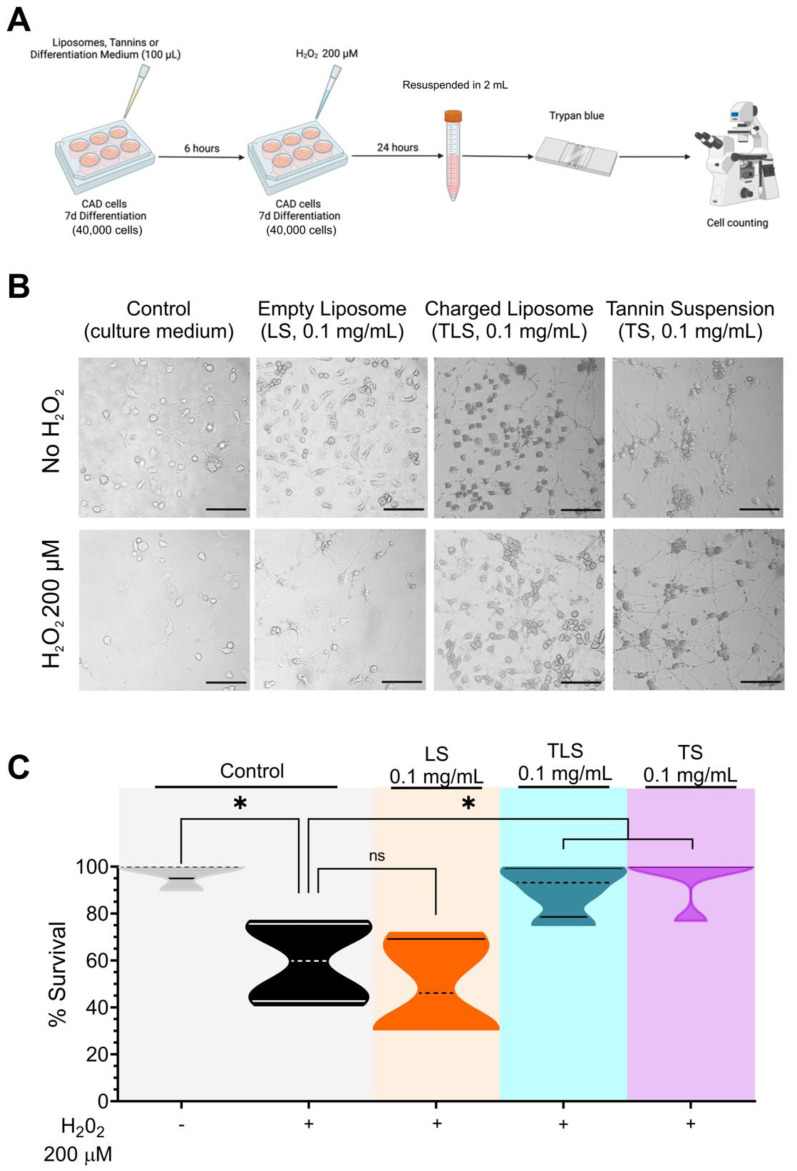
Effect of lipid-encapsulated grape tannins on neuronal cell survival under oxidative stress. (**A**) Experimental design schematization. (**B**) Representative image (20X) of each condition, taken in a phase contrast microscope. Scale bars indicate 50 µm. (**C**) Cell survival normalized to Control plates live cells number. Control, H_2_O_2_, LS, TLS (n = 4), TS (n = 5). Each n represents a different cell plate from distinct batches and all measurements were performed in quadruplicate in a different experiment day. One-way ANOVA followed by Bonferroni’s multiple comparisons test. *: *p* < 0.05. Normal distribution was assessed by Shapiro–Wilk test (W values 0.8634; 0.9975; 0.8814; 0.9643; 0.9094; Control, H_2_O_2_, LS, TLS and TS, respectively; *p* > 0.05).

**Figure 2 antioxidants-11-01928-f002:**
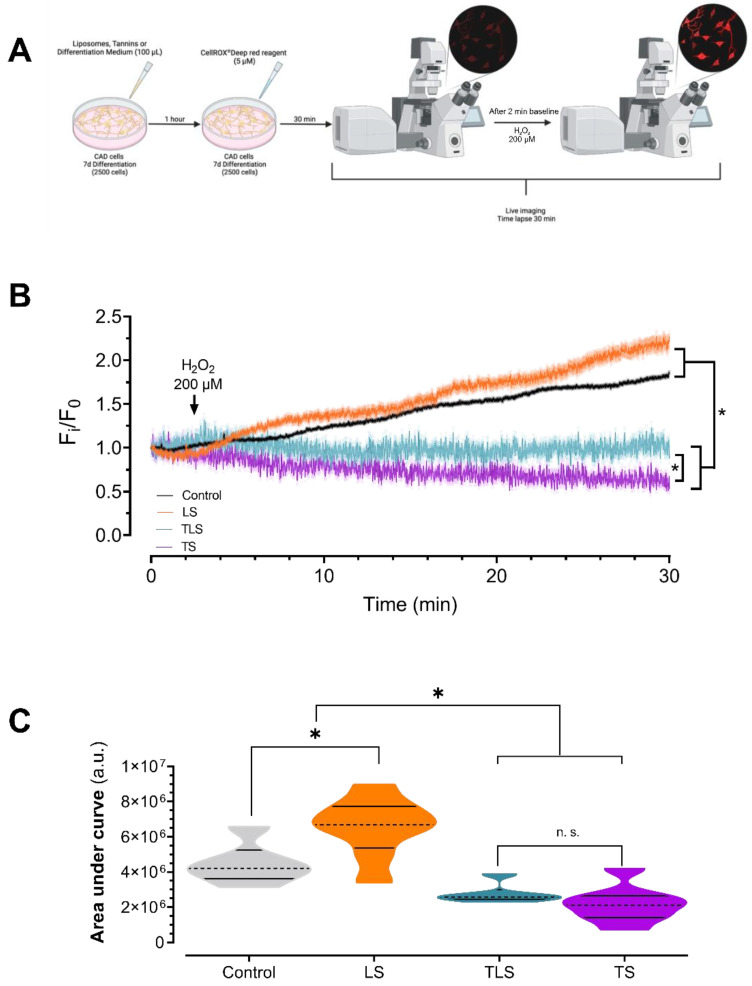
Effect of lipid-encapsulated grape tannins on intracellular reactive oxidative species levels. (**A**) Schematization of experimental procedures. (**B**) Representative time-lapse CellROX experiments normalized by initial fluorescence, after exposition to H_2_O_2_. Two-way ANOVA and repeated measurements. *: *p* < 0.05. (**C**) Integrated fluorescence response induced by H_2_O_2_. One-way ANOVA followed by Bonferroni’s post hoc test, *: *p* < 0.05. Control (n = 47 cells), LS (n = 65 cells), TLS (n = 51 cells) and TS (n = 47 cells). Normal distribution was assessed by Shapiro–Wilk test (W values 0.8634; 0.9975; 0.8814; 0.9643; 0.9094; Control, H_2_O_2_, LS, TLS and TS, respectively; *p* > 0.05).

**Figure 3 antioxidants-11-01928-f003:**
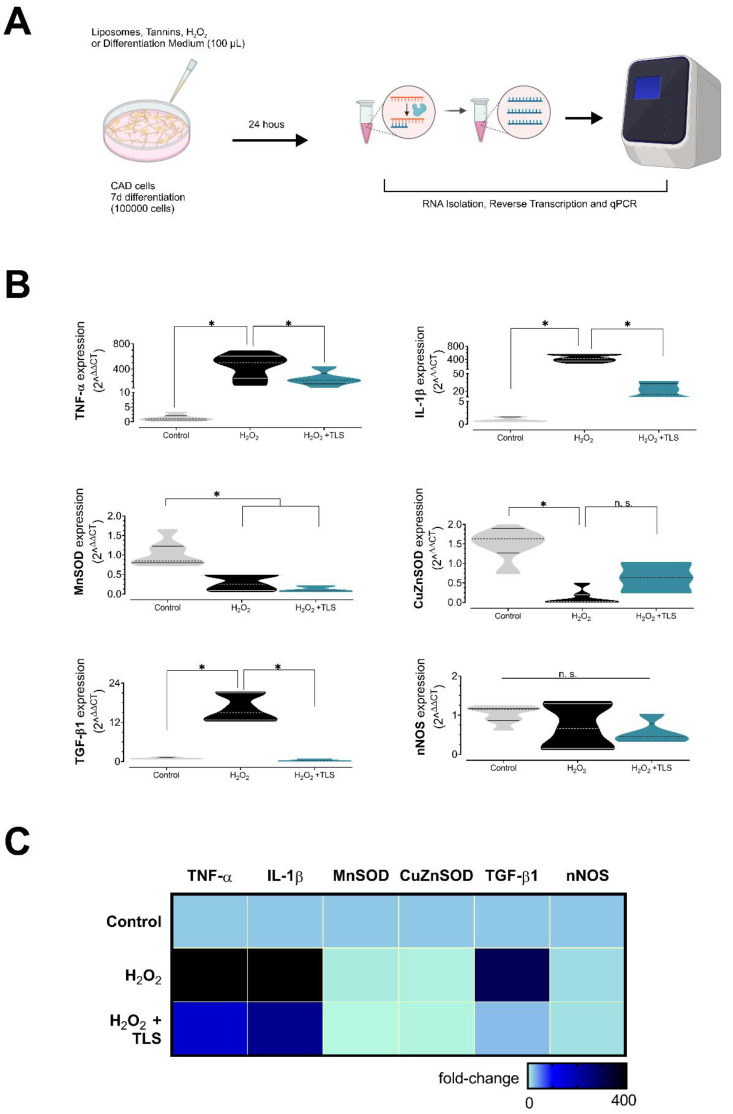
Effect of lipid-encapsulated grape tannins on neuroinflammatory biomarkers. (**A**) Schematization of experimental procedures. (**B**) Results of qPCR by the 2^ΔΔCT^ method showing mRNA expression of proinflammatory biomarkers, antioxidant enzymes and anti-inflammatory biomarkers. (**C**) Heat map showing the average mRNA expression (geometric mean) of gene targets evaluated in (**B**). n = 5 per group, on which each n represents a different batch of CAD cells. One-way ANOVA followed by Bonferroni’s post hoc test. *: *p* < 0.05. Normal distribution was assessed by Shapiro–Wilk test (W values 0.8882; 0.9513; 0.8503; Control, H_2_O_2_, and H_2_O_2_ + TLS, respectively; *p* > 0.05).

**Table 1 antioxidants-11-01928-t001:** Lipid-encapsulated grape tannins characterization.

Analysis	TS	TLS
*Particle size* Mean particle size (MPS) Polydispersity index (PDI) Z-potential (ζ) *Proximal Analysis*	742.7 ± 5.30 nm 0.67 ± 0.03 −13.50 ± 1.70 mV Content (g/100 g)	309.9 ± 5.90 nm 0.41 ± 0.02 −21.40 ± 1.70 mV Content (g/100 g)
Moisture Ash	95.06 0.57	96.64 0.57
Protein	0.08	0.08
Lipids	0.48	0.18
N.N.E	2.64	2.53
Kcal/100 g	11.39	12.04
	GAE (µg/mL)	GAE (µg/mL)
*Total polyphenols*	78.34 ± 0.12 *	30.23 ± 0.14
	TEAC (µmol TE/mL)	TEAC (µmol TE/mL)
*Antioxidant activity (ABTS)*	2017.51 ± 238.57 *	9.10 ± 6.75

Data are shown as mean ± S.D. N.N.E: Non-nitrogenated extracts. GAE: Gallic acid equivalents expressed as µg of gallic acid per mL of extract. TEAC: Antioxidant capacity expressed as µM of Trolox equivalents per mL. Total polyphenols and ABTS assays were performed in triplicate in five independent experiments. Unpaired *t*-tests, * *p* < 0.05 vs. TLS.

## Data Availability

The data presented in this study are available within the article and supplementary material.

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
