# Peer review of "Lipid-Encapsuled Grape Tannins Prevent Oxidative-Stress-Induced Neuronal Cell Death, Intracellular ROS Accumulation and Inflammation"

_antioxidants, 2022, doi:10.3390/antiox11101928_

Round 1
Reviewer 1 Report
The paper of Diaz et al investigates the effect of lipid encapsulated tannin in CAD cells compared to tannin alone or solvent control. Main findings are that encapsulated tannins are able to protect CADS cells from H2O2 induced cell death, have positive effects in respect to inflammation and reactive oxidative species. Methods are mainly based on cell viability assays, RT-PCR (deltadelta CT method) and a ROS assay. In principle I fully agree to the authors that the main problem with polyphenols or other natural compound is the low bioavailability. Therefore, enhancing the bioavailability with e.g. lipid encapsulation is a very interesting and promising technique and the result is interesting for a broad readership and in my opinion in the scope of the journal.
However, I have some major points which needs to be addressed before the paper is suitable for publication in my opinion:
Principal concerns:
- The paper is based on pure cell culture experiments. To transfer these results in vivo is at this point not possible. Although these data are very interesting, the authors completely overinterpret their data. How does lipid encapsulation described here affect the transport of tannins over the gut barrier or even worse the blood brain barrier. In vivo experiments are needed to confirm the results.
- The statistic is not described sufficiently. Which statistical method was used for which experiments? Which test was used to check for normal distribution etc.
Major concerns:
- The tannin concentration seems very high in these experiments? What are the concentrations expected to be in the brain after consumption? Are these physiological concentrations?
- The techniques used here are just scratching the surface of the mechanisms of inflammation. The authors should discuss their results much more carefully.
- Only 1 house keeping gene is used for normalization in the delta delta CT RT-PCR method. No information is provided if this gene is stable under the incubation conditions used in this paper. Please test experimentally if this gene is stable and therefore suitable for normalization.
Minor concerns:
- Please provide some information about the Phosphatidylcholine used here. What is the fatty acid composition?
Reviewer 2 Report
It is opinion of the reviewer that this paper before acceptance needs several corrections. My individual comments are listed below.
The title should be written with first capital letters.
L. 7-15 – The authors’ initials should be added.
L. 49 – Remove one “chronic”.
L. 56 – It should be “Toll-like Receptors”.
L. 70 – It should be “chelation of prooxidant metal ions”’.
In the Introduction, applications of liposomes as phenolic vehicles should be better described.
L. 136 – What was a sample?
L. 140 – It should be “HPLC-UV”
L. 143 – It should be “(acid thiolysis)”.
The acid thiolysis should be nriefly described.
The condition of HPLC must be described,
The results of HPLC analysis must be described.
L. 145 – It should be “…antioxidant …”.
L. 155 – It should be “ … Trolox equivalents (TE) … (μmol TE/mL)”.
L. 194 – It should be “…microplate reader”. Its model should reported.
L. 223 – It should be “t-Student test”.
L. 223 – A Bonferroni’s post-hoc test must be mentioned.
L. 230 – It should be “lipids” instead of “fat”.
L. 235 , Table 1– It should be “μmol TE/mL”.
Table 1 – It should be “… Moisture …Ash … Protein ”
Table 1 – Kcal per ???
Table 1 – The low TEAC of TLS needs to be discussed.
L. 327, 362 – Condensed tannins are typical for grape seeds, not hydrolyzable ones (tannic acid).
L. 338 – It should be “did not investigate”.
Reference – Abbreviations of all journal titles are needed.
Reviewer 3 Report
Several significant drawbacks have impacted the methodology, the results, and the conclusion of your study:
1. The tannic acid used in this study was purchased from sigma, I don’t think it would be proper to use "grape tannins" in the title.
2. Current knowledge of tannin-encapsulated liposome, especially those studies related to neuroprotection should be provided.
3. Numerous typographical mistakes and grammatical errors are noticed. For example: Line 234: “TLS showed negligible antioxidant activity compared to TLS”??
4. What are the drug loading percentage and encapsulation efficiency? How the concentrations of TLS and TS used in Fig 1 been determined? What were the concentrations of TLS and TS used in Fig 2 and Fig 3?
5. Please explain why total phenol content and antioxidant activity of TLS were lower than that of TS? If the encapsulated tannic acid was re-extracted from TLS, whether they still show antioxidant activity?
6. There were 8 groups in Fig 1B, but only 5 groups in Fig 1C, please explain.
7. What is “charged LS”?
8. Line 252: “empty liposomes were unable to prevent cell death after H2O2 treatment”, however, according to the picture showed in Fig 1B, the H2O2/LS group looked very similar to H2O2/TS group, bur very different from the H2O2/Control group. And why the cell number in “No H2O2/LS” group looked much fewer than other “No H2O2” treatment groups?
9. In Fig 2B the line of LS group was below that of the control group, however, in Fig 2C, the area under curve of LS group was significantly higher than that of the control group, please explain.
10. There was no difference between TLS and TS group in Fig 1C and 2C. Then how the authors can conclude that “liposomes efficiently delivered tannins”?
11. There was no citation for Fig 1A, 2A, and 3A.
12. What was the purpose to re-present the results of Fig 3B in a different way in Fig 3C?
13. Why did the LS and TS group not be compared in Fig 3?
14. There is an unjustified statement that is neither concluded nor validated by the results in this preliminary in-vitro study: “lipid encapsulation of tannins resulted in effective delivery of grape seed polyphenols to neurons, and promoted neuroprotection BY A DUAL MECHANISM including direct ROS scavenging and inhibition of neuronal inflammatory response”
15. There was no result related to bioavailability.
Round 2
Reviewer 1 Report
The authors have addressed all of my concerns sufficiently. Data is now presented more carefully and statistic is described adequately, I therefore recommend publishing the study in its present form.
Author Response
We thanks reviewer 1 for his/her recommendation of acceptance.
Reviewer 2 Report
The authors corrected this paper properly taken under considerations all my comments. Therefore, I can accept it now.
Author Response
We thanks reviewer 2 for his/her recommendation of acceptance
Reviewer 3 Report
1. The answer of Question #10 should be added into discussion section.
2. To directly show that liposome formulation does have benefit in protecting neuron damage, comparative studies on survival and ROS level between TLS and equivalent amount of TS treated cells should be performed.
Author Response
- The answer of Question #10 should be added into discussion section.
We agree with reviewer comment and decided to include the answer to the previous comment (Question#10 R1) on the discussion section of our revised manuscript as requested by the reviewer.
- To directly show that liposome formulation does have benefit in protecting neuron damage, comparative studies on survival and ROS level between TLS and equivalent amount of TS treated cells should be performed.
We partially agree with reviewer comment. However, our data showed that TLS showed a 2-fold more in vivo antioxidant capacity compared to TS. Indeed, we found that TLS containing 0,05 mg/mL of tannins mimicked the effects of TS at 0,1 mg/mL. Therefore, while we think the experiments suggested by the reviewer could be done in future studies, we feel that our data showing that TLS are more potent means to deliver tannins into neurons in vitro compared to TS fits the main aim of the present study that was to provide evidence about TLS being efficient to reduce neuronal cell death induced by an oxidant insult.